# Investigating Snake-Venom-Induced Dermonecrosis and Inflammation Using an Ex Vivo Human Skin Model

**DOI:** 10.3390/toxins16060276

**Published:** 2024-06-17

**Authors:** Jaffer Alsolaiss, Gail Leeming, Rachael Da Silva, Nessrin Alomran, Nicholas R. Casewell, Abdulrazaq G. Habib, Robert A. Harrison, Cassandra M. Modahl

**Affiliations:** 1Centre for Snakebite Research and Interventions, Liverpool School of Tropical Medicine, Liverpool L3 5QA, UK; rachael.da-silva@lstmed.ac.uk (R.D.S.); noalomran@moh.gov.sa (N.A.); nicholas.casewell@lstmed.ac.uk (N.R.C.); robert.harrison@lstmed.ac.uk (R.A.H.); cassandra.modahl@lstmed.ac.uk (C.M.M.); 2Abqaiq General Hospital, Rural Health Network, Eastern Health Cluster, Ministry of Health, Abqaiq 33241, Saudi Arabia; 3Department of Veterinary Anatomy, Physiology and Pathology, School of Veterinary Science, University of Liverpool, Liverpool L69 7ZX, UK; gleeming@liverpool.ac.uk; 4Qatif Medical Fitness Center, Clinical Laboratory Department, Qatif Health Network, Eastern Health Cluster, Ministry of Health, Qatif 31911, Saudi Arabia; 5African Snakebite Research Group (ASRG) Project, Bayero University, Kano 700251, Nigeria; abdulrazaq_habib@yahoo.co.uk

**Keywords:** viper, elapid, cytotoxicity, blistering, immunology, snakebite envenoming therapies

## Abstract

Snakebite envenoming is a neglected tropical disease that causes >100,000 deaths and >400,000 cases of morbidity annually. Despite the use of mouse models, severe local envenoming, defined by morbidity-causing local tissue necrosis, remains poorly understood, and human-tissue responses are ill-defined. Here, for the first time, an ex vivo, non-perfused human skin model was used to investigate temporal histopathological and immunological changes following subcutaneous injections of venoms from medically important African vipers (*Echis ocellatus* and *Bitis arietans*) and cobras (*Naja nigricollis* and *N. haje*). Histological analysis of venom-injected ex vivo human skin biopsies revealed morphological changes in the epidermis (ballooning degeneration, erosion, and ulceration) comparable to clinical signs of local envenoming. Immunostaining of these biopsies confirmed cell apoptosis consistent with the onset of necrosis. RNA sequencing, multiplex bead arrays, and ELISAs demonstrated that venom-injected human skin biopsies exhibited higher rates of transcription and expression of chemokines (CXCL5, MIP1-ALPHA, RANTES, MCP-1, and MIG), cytokines (IL-1β, IL-1RA, G-CSF/CSF-3, and GM-CSF), and growth factors (VEGF-A, FGF, and HGF) in comparison to non-injected biopsies. To investigate the efficacy of antivenom, SAIMR Echis monovalent or SAIMR polyvalent antivenom was injected one hour following *E. ocellatus* or *N. nigricollis* venom treatment, respectively, and although antivenom did not prevent venom-induced dermal tissue damage, it did reduce all pro-inflammatory chemokines, cytokines, and growth factors to normal levels after 48 h. This ex vivo skin model could be useful for studies evaluating the progression of local envenoming and the efficacy of snakebite treatments.

## 1. Introduction

Annually, up to 138,000 individuals die and over 400,000 individuals suffer life-altering disabilities as a result of snakebite envenoming, with the highest rates of incidence disproportionally affecting impoverished inhabitants of rural tropical regions [1,2]. Consequently, snakebite envenoming is a World Health Organization-listed neglected tropical disease. Due to the numerous different snake species responsible for envenoming and the extensive variation in their venom composition [3], the pathophysiology following envenoming can vary extensively, with systemically envenomed victims presenting with signs of neurotoxicity (e.g., ptosis and respiratory paralysis) and/or haemotoxicity (e.g., haemorrhage, coagulopathy, and hypotension), both of which can be life-threatening [2]. Symptoms of local envenoming include instant radiating pain, tender swelling that expands rapidly, inflammatory erythema, lymph-vessel inflammation manifesting as red lines on the skin (lymphangitis), protracted haemorrhage from bite puncture wounds, blistering (bullae), bruising (ecchymosis), increases in the sizes of local lymph nodes, and/or necrosis of soft tissue and muscle [2,4,5].

Viperid venoms and venoms from some elapid species, such as spitting cobras, cause rapid and extensive local tissue damage [2,6,7], as well as local pain or hyperalgesia due to inflammation [8,9,10,11]. In viperids, this is primarily due to snake venom metalloproteinases (SVMPs) degrading cell basement membranes at the dermal–epidermal interface, separating the dermis from the epidermis, and forming blisters [12]. In addition to basement membrane hydrolysis, SVMPs cause damage to microvasculature and widespread degradation of the extracellular matrix, resulting in haemorrhage, tissue hypoxia, and impaired regeneration [13]. Viperid group II phospholipase A_2_s (PLA_2_s) have been found to contribute to the accumulation of fluid in the tissue (oedema) by inducing cell contraction, leading to irreversible cell damage [14]. In elapid venoms, cytotoxic PLA_2_s and three-finger toxins directly damage cells, disrupting cell membranes and/or inducing pore formation, including lysosome lysis [15,16,17]. For both viperid and elapid venoms, SVMPs and PLA_2_ promote pro-inflammatory responses that could mediate venom-induced tissue damage, however the exact contributions of toxins and inflammation to acute and longer-term local tissue lesions remain unclear [18]. 

Our current knowledge of local envenoming pathogenesis emanates primarily from subjecting murine models to venoms from Latin American snakes. For example, an isolated P-I SVMP from *Bothrops asper* venom induced paw oedema and blister formation, in addition to increases in inflammatory mediators (e.g., IL-1 and IL-6) and leukocyte adhesion molecules [19,20,21,22]. Blistering likely develops from protein hydrolysis at the interface between the dermis and epidermis, as suggested by the dermoepidermal division, alongside the occurrence of basement membrane protein fragments in exudates gathered after skin sectioning in mouse models [19,23]. Large-scale local inflammatory processes triggered by envenoming include synthesis and release of eicosanoids, nitric oxide (NO), bradykinin, complement anaphylatoxins, histamine and cytokines (interleukins IL-1β, IL-6, and IL-10 and tumour necrosis factor α), activation of resident macrophages and other inflammatory cells, and mobilisation of leukocytes and neutrophils [24,25,26,27,28,29]. Increased vascular permeability results in the development of an exudate that contains plasma proteins, alongside intracellular and extracellular protein fragments, chemokines, cytokines, and damage-associated molecular patterns (DAMPs), likely aggravating inflammation and intensifying tissue damage [30,31,32,33]. There is also limited knowledge of the local envenoming pathology associated with African vipers and spitting cobras, with what is currently known primarily being derived from limited clinical observations [7,34]. 

While murine model studies have provided important information on the systemic and local effects of snake venoms in vivo [35], there remain considerable ethical and financial concerns with such assays, which cause severe pain, harm, and distress to experimental animals. There is therefore a compelling 3R (‘Replace, reduce and refine the use of experimental animals in research.’) rationale for finding alternatives to these in vivo murine tests [36]. Furthermore, there are concerns over the real-world applicability of such animal models [37], in particular the extent to which pathophysiological responses of venom-injected mice are accurate representations of human envenoming [38]. Several in vitro assays [39,40,41] have been used to identify and assess venom toxins’ function and interaction with existing and promising snakebite therapies; these have included cell-based assays used to gain better insight into the tissue damage relevant to human snakebite victims [42,43,44]. However, there are still currently no alternate models for in vivo testing that accurately recapitulate the effects of local envenoming with the complete functional, structural, and biological complexity of skin, including local pro-inflammatory responses.

To address this, we tested the utility of an ex vivo human skin model to better understand human-tissue envenoming from African vipers and cobras. This was achieved by investigating the local tissue damage and pro-inflammatory responses of human skin biopsies subcutaneously injected with venoms from African saw-scaled vipers (*Echis ocellatus*), puff adders (*Bitis arietans*), and spitting and non-spitting cobras (*Naja nigricollis* and *N. haje*, respectively), species whose envenoming encapsulates the diverse systemic and local pathologies observed in African snakebite victims. Our findings suggest that an ex vivo skin model could be developed that reduces our current dependency on in vivo mouse models for the study of local envenoming, while simultaneously providing a human model of greater relevance for the treatment of snakebite victims.

## 2. Results

### 2.1. Human HypoSkin^®^ Morphology Seven Days following Envenoming

The human HypoSkin^®^ model was macro- and microscopically evaluated for venom-induced lesions, ulceration, and sloughing/loss of the epidermis. A progressive series of changes were macroscopically observed in skin injected with *E. ocellatus*, *B. arietans*, and *N. nigricollis* venom. The temporal development of a faint brownish lesion surrounded by a whitish area in the epidermis and/or superficial dermis at seven days (168 h) was most pronounced in the biopsy injected with *B. arietans* venom (Figure 1). Both the ‘non-necrotic’ *N. haje* venom-injected and the non-injected control biopsy exhibited an unaltered macroscopic appearance (Figure 1).

Histological analysis of venom-injected biopsies at day seven showed evidence of venom-induced pathological alterations, in comparison to the untreated controls. For the *E. ocellatus* and *B. arietans* viperid venoms, there was loss of skin viability, evidenced by a separation of the epidermis from the dermis (Figure 2A), and a ballooning degeneration of the epidermal cells. Pyknotic nuclei, potentially indicating apoptotic or necrotic cells, were observed in epidermal crests as well as in the deep dermis for all venom treated biopsies; this determination included pyknotic nuclei observed in the dermis in the *N. haje* venom-injected biopsy (Figure 2A). The non-injected control biopsies each had an unaltered epidermis and dermis, including skin appendages (Figure 2A).

### 2.2. Extent of Cell Apoptosis and Necrosis Seven Days following Envenoming

To confirm the presence of apoptotic cells, skin biopsies were stained with an active apoptotic caspase-3 antibody and terminal deoxynucleotidyl transferase dUTP nick end labelling (TUNEL). Apoptotic cells were evident in all envenomed biopsies, including the neurotoxic control venom from *N. haje* (Figure 2, see panels 5B and 5C). The active caspase-3 antibody positively stained locations where cells exhibited morphological characteristics of apoptosis in addition to the evidence of the translocation of caspase-3 to the nucleus which is suggestive of the progression of apoptosis. Caspase-3 staining was also apparent in unaltered cells, which would indicate the early stages of apoptosis [45]. Cell mortality was observed microscopically in all venom-injected biopsies, albeit with different levels of TUNEL-positive cells (Figure 2 and Figure 3). The TUNEL staining results thus confirm the presence (Figure 2 and Figure 3) of apoptotic cells in all biopsies and the presence of necrotic cells in biopsies injected with venoms from vipers *E. ocellatus* and *B. arietans* (significant increases vs. untreated control, *p* = 0.0003 and 0.0002, respectively). Overall, the largest numbers of apoptotic and necrotic cells were observed in biopsies treated with *B. arietans* venom (significant increase vs. untreated control, *p* < 0.0001), and lower numbers of apoptotic cells were observed in biopsies treated with *N. nigricollis* or *N. haje* venom (no significant difference vs. untreated control). Importantly, no evidence of apoptosis was observed in untreated controls or the PBS-injected control.

### 2.3. Human HypoSkin^®^ Histology 48 h following Envenoming and Antivenom Treatments

To evaluate the early progression of lesion development and dermonecrosis, histological examination was performed on skin biopsies 6, 12, 24, 36, and 48 h after subcutaneous venom injection. Focal changes to the epidermis, consisting of varying levels of (i) degeneration (ballooning degeneration, pyknosis, and hypereosinophilia) and (ii) loss of the epidermis (erosion and/or ulceration) were evident in all of the venom-injected biopsies from 24 h (Figure 4 and Appendix A). Biopsies envenomed with *E. ocellatus* showed evidence of hypereosinophilia and pyknosis 24 h post-envenoming but were unremarkable at earlier and later time points. Multifocal epidermal loss was evident in biopsies envenomed with *B. arietans* at 36 h post-envenoming, and ulceration, pyknotic nuclei, and hypereosinophilia were evident at 48 h. Biopsies envenomed with *N. nigricollis* did not exhibit any remarkable changes, and biopsies envenomed with *N. haje* showed mild multifocal erosion and ulceration of the epidermis only at 48 h. However, there was considerable biopsy variation, even when the same venom was injected, with later time points not always showing similar progressions of alterations in the epidermis. 

Antivenom interventions, which were delivered one hour following the subcutaneous injection of *E. ocellatus* or *N. nigricollis* venom into the biopsies, were evaluated at 48 h to determine whether antivenom treatments were effective in neutralising the venom-induced pathologies. Envenomed biopsies that were subcutaneously injected with antivenom did not exhibit reductions in the degeneration or loss of the epidermis (Figure 5). Focal loss of the epidermis with ballooning degeneration and apoptotic cells at the edge of the ulceration were apparent for the *E. ocellatus* venom and SAIMR Echis monovalent antivenom-treated biopsy (Figure 4A,B and Appendix A). The biopsy injected with *N. nigricollis* venom, followed after one hour by SAIMR polyvalent antivenom, did not show any remarkable change, which was consistent with the *N. nigricollis* venom-only biopsies over the 48 h time course (Figure 4 and Figure 5C,D). 

### 2.4. Expression of Pro-Inflammatory Chemokines, Cytokines, and Growth Factors following HypoSkin^®^ Envenoming and Treatments

Pro-inflammatory processes arising from local tissue damage in biopsies following venom injections and after antivenom interventions were quantified by RNA sequencing (Appendix A). The neutrophil-recruiting C-X-C motif chemokine ligand 5 (*CXCL5*) was identified among the top 50 upregulated genes at 48 h for all venom-injected biopsies (Appendix A). Of the top 50 upregulated genes at 48 h, skin biopsies that had been injected with necrotic venoms (*E. ocellatus*, *B. arietans*, and *N. nigricollis*), exhibited expression of three common genes: keratin 23 (*KRT23*), secreted phosphoprotein 1 (*SPP1*), and TIMP metallopeptidase inhibitor 1 (*TIMP1*) (Appendix A). KRT23 is responsible for the structural integrity of epithelial cells. SPP1 binds to integrin receptors on leukocytes to induce adhesion, migration, and survival, and functions as a cytokine upregulating the expression of IFN-γ and IL-12. TIMP1 is an inhibitor of matrix- and disintegrin-metalloproteinases [46]. 

Four genes, keratin 23 (*KRT23*), a pseudogene for complement component 1 Q subcomponent-binding protein (*C1QBPP2*), aldehyde dehydrogenase 3 family member B2 (*ALDH3B2*), and ankyrin 3 (*ANK3*), were upregulated in *E. ocellatus* venom-injected biopsies for all time points (Appendix A). Of these, C1QBP has been identified as having a pro-inflammatory role, promoting the migration of macrophages [47]. Gene ontology annotations for genes upregulated 2-fold or greater were enriched for skin development, keratinocyte differentiation, and keratinisation at 48 h in *E. ocellatus* venom-injected biopsies (Appendix A).

Multiple S100 genes (*S100A7*, *S100A8*, and *S100A9*), *KRT23*, and histidine ammonia-lyase (*HAL*) were upregulated at all time points after *B. arietans* venom injection (Appendix A). Genes upregulated over 2-fold were also enriched for tissue development in *N. nigricollis* venom-injected biopsies at 48 h, and this upregulation was seen from 24 h (Appendix A). Genes associated with tissue development were enriched at 48 h following *B. arietans* venom injection (Appendix A).

Four genes related to post-translational protein processing were upregulated at all time points in *N. nigricollis* venom-injected biopsies (Appendix A); these included two genes for the signal peptide recognition particle RNA (*SRP*) and pseudogenes for 7SL RNA (*RN7SL51P* and *RN7SL190P*), components of the signal-recognition particle. For the *N. haje* venom-injected skin biopsy, 33 of the upregulated genes were not common to the necrotic venom-injected biopsies at 48 h, however there was overlap in the upregulated gene expression profiles of *N. haje* and the necrotic venoms (Appendix A). This overlap included genes for S100 proteins (*S100A7*, *S100A8*, and *S1009*), among which *S100A7* was upregulated at all *N. haje* time points (Appendix A), and genes involved in keratinocyte differentiation, where this biological process was upregulated more than 2-fold as early as 12 h following venom injection. 

To determine alterations in gene expression after antivenom interventions, expression levels of the top 50 upregulated genes 48 h after venom treatments were evaluated in antivenom-treated biopsies. It was found that after antivenom interventions, *CXCL5* upregulation was reduced in biopsies that had been injected with venom, and the three genes observed to be commonly upregulated in skin biopsies injected with necrotic venoms (*KRT23*, *SPP1*, and *TIMP1*) were no longer present among the top 50 upregulated genes at 48 h following venom injection (Appendix A).

Gene expression corresponding to the pro-inflammatory proteins screened in the Human Cytokine Magnetic 30-Plex Panel (Figure 6A) identified increases in expression over 2-fold for chemokine ligand 3 (*CCL3*), ligand 5 (*CCL5*), granulocyte-macrophage colony-stimulating factor (*CSF2*), CXC chemokine ligand 8 (*CXCL8*), hepatocyte growth factor (*HGF*), interleukin 1 beta (*IL-1β*), interleukin 1 receptor antagonist *(IL1RN*), interleukin 7 (*IL-7*), tumour necrosis factor (*TNF*), and vascular endothelial growth factor A (*VEGF-A*). *VEGF-A* was upregulated after *E. ocellatus* venom injection from six hours (the earliest time point evaluated) to 48 h (the latest time point), and after *B. arietans* venom injection from six to 24 h, and again at 48 h. In addition, *VEGF-A* upregulation was seen after *N. haje* venom injection at six hours. *CXCL8* and *IL-1β* were upregulated at multiple time points, primarily after *Naja* venom treatments. Gene expression of *IL-6* was found to be downregulated in our dataset for all venoms, although the extent of downregulation varied across venoms and time points. 

Skin biopsies injected with SAIMR Echis monovalent or SAMIR polyvalent antivenom 48 h following *E. ocellatus* and *N. nigricollis* venom injections, respectively, had reduced expression levels of *CCL3*, *CXCL8*, *IL-1β*, *IL1RN*, *IL-7*, *TNF*, and *VEGF-A*, compared to biopsies injected with venom only. However, *CCL5* expression levels increased after antivenom interventions.

### 2.5. Secretion of Pro-Inflammatory Chemokines, Cytokines, and Growth Factors following HypoSkin^®^ Envenoming and Treatments

Changes in the levels of four chemokines (MIP1-ALPHA, RANTES, MCP-1, and MIG), five cytokines, (IL-1β, IL-1RA, IL-12, G-CSF/CSF-3, and GM-CSF), and three growth factors (VEGF-A, FGF-Basic, and HGF) were the most notable in the cell-culture media collected from biopsies following venom injections over the 120 h time course (Figure 6B and Figure 7, Appendix A). Biopsies injected with *E. ocellatus*, *B. arietans*, and *N. haje* venom exhibited peak levels of the chemokine Macrophage Inflammatory Protein-1α (MIP1-ALPHA) at 72 h (Figure 7), followed by a gradual reversal to normal by 120 h. In biopsies injected with *N. nigricollis* venom, MIP1-ALPHA increased at 36 h and reverted to normal levels by 72 h. Regulated-upon-activation, normal T cell-expressed, and secreted (RANTES) levels were highest in media from *E. ocellatus* venom-injected biopsies, peaking at 96 h. Monocyte Chemoattractant Protein-1 (MCP-1) increases were most notable after 24 h post-subcutaneous *N. nigricollis* envenoming, with both MIP1-ALPHA and MCP-1 peaking at 48 h, before normalising. The C-X-C chemokine Monokine Induced by Gamma interferon (MIG) was elevated 24 h post-*B. arietans*-envenoming but decreased after this time point, and this peak was only present in *B. arietans* biopsy media.

For the cytokines, one of the most apparent inflammatory responses was a rapid significant increase in IL-1β over 48 h for biopsies injected with *E. ocellatus* (*p* < 0.0001 vs. untreated control) and *B. arietans* venoms, which then declined by 96 h, while biopsies injected with the venom of *N. nigricollis* and *N. haje* had incremental increases in IL-1β in the first 24 h, before reverting to normal by 72 h (Figure 7). Levels of IL-1RA generally increased in the later stages of the experiments, and for the viper venoms, increases occurred after 72 h for *E. ocellatus* and 96 h for *B. arietans*, with the latter increasing again most markedly after 120 h. For the elapid venoms, IL-1RA was significantly increased at 48 h (*p* < 0.0001 vs. untreated control), and thereafter remained stable for *N. haje* but continued to increase for *N. nigricollis*. IL-12 levels were higher than the control for biopsies injected with venom from 6 h onward, but this upward tread notably increased in *E. ocellatus* and *B. arietans* venom-injected biopsies after 96 h. *N. nigricollis* and *N. haje* envenoming elevated Granulocyte Colony Stimulating Factor (G-CSF) and Granulocyte-Macrophage Colony-Stimulating Factor (GM-CSF) within 24–48 h (*p* < 0.0001 vs. untreated control), and at later time points for the viper venoms, continuing to increase for *E. ocellatus* and *B. arietans* up to, and likely past, 120 h (*p* < 0.0001 vs. untreated control).

For growth factors, viper envenoming of HypoSkin^®^ tissue elevated Vascular Endothelial Growth Factor A (VEGF-A), Fibroblast Growth Factors (FGF-basic), and Hepatocyte Growth Factor (HGF) in biopsy culture media after 24 h. Injections with *E. ocellatus*, *B. arietans*, and *N. nigricollis* venoms led to progressive increases in VEGF-A after 48 h, peaking at 96 h, and reverted to control levels by 120 h. By contrast, injection with *N. haje* venom did not trigger any marked elevations in VEGF-A. Culture media from *E. ocellatus* and *B. arietans* venom-injected biopsies showed elevated FGF at 24 h, which then decreased to normal levels at 120 h. The levels of HGF remained stable throughout the experiment, except for a substantial increase at 48 h in the biopsies injected with *B. arietans* venom, and a much more minor increase seen at the same time point from *N. nigricollis* venom (Figure 7). In all cases, the levels of pro-inflammatory chemokines, cytokines, and growth factors reduced to normal levels after 48 h following injection of antivenom (SAIMR Echis monovalent or SAMIR polyvalent), in comparison to samples collected from skin models with venom-only injections.

## 3. Discussion

We instigated this study to better understand the pathology and the role of inflammatory processes in the pathogenesis of the local, tissue-destructive effects of African snake envenoming. To that end, for the first time, an ex vivo human skin model was used instead of the more commonly used murine in vivo model, because of the previously outlined ethical and human–mouse physiological concerns. We used venoms from sub-Saharan African vipers (*E. ocellatus* and *B. arietans*) and a spitting cobra (*N. nigricollis*) as test vehicles because, taken together, their envenoming is responsible for a considerable amount of snakebite morbidity. Whereas the murine model deploys an intradermal route of venom injection, we subcutaneously injected the test venoms into ex vivo human skin biopsies to better reflect human envenoming. Our results therefore provide insight into venom action and human-tissue pro-inflammatory responses to snakebite, which will aid in the identification and design of therapeutics that could be of use to mitigate venom-induced tissue damage, as well as demonstrate how an ex vivo human skin model can be applied to evaluate the progression of local envenoming and the efficacy of snakebite treatments.

We were concerned that the non-perfused nature of the ex vivo human skin biopsies might severely limit the utility of this model, especially for multi-day experiments. It was gratifying, therefore, to observe that the non-venom-injected controls showed no histopathological, apoptotic, or immunological evidence of tissue degeneration over the seven-day (168 h) duration of our experiments. Contrastingly, the human skin biopsies showed evidence of faint brownish lesions after subcutaneous injections of necrotic venoms from *E. ocellatus*, *B. arietans*, and *N. nigricollis*. These lesions were not as distinctive as those which are observed in mouse dermonecrosis models (i.e., MND), even though the doses used were three times greater than mouse MNDs. This is likely due to (i) our use of subcutaneous venom injection rather than the intradermal route used in the murine model, (ii) the anatomical differences between human skin and mouse skin (e.g., significantly thicker epidermis and dermis in humans) [48], and (iii) the non-vascularised nature of the human biopsies, all factors that likely contribute to the distribution of toxins and their tissue-destructive effects. Histological analysis of the skin biopsies identified ballooning degeneration of epidermal cells, separation of the epidermis from the dermis after subcutaneous injections of viperid venoms *E. ocellatus* and *B. arietans*, and hypereosinophilia and pyknosis within 24 h for *E. ocellatus* and 36 h for *B. arietans.* Biopsies subcutaneously injected with *N. nigricollis* venom did not exhibit any remarkable change over the 168 h time-course, although pyknotic nuclei were detected. Mice injected intradermally with *N. nigricollis* venom exhibit dermal oedema, blistering, loss of skin appendage, cellularity reduction, and dermonecrosis within 24 h, which has been associated with epidermal loss and replacement by fibrinoid hyaline material [49]. Light micrograph sections of those mouse skins showed thrombi formation in blood vessels, and ischaemia associated with thrombosis and vascular injury, leading to dermonecrosis, from *N. nigricollis* venom in the mouse model [49]. Furthermore, changes such as oedema can only occur in vascularised tissue. It is therefore the lack of reliance on vascular circulation for nutrients (due to provision via cell-culture medium) as well as the reduced ability to induce a vascular inflammatory response in the human skin biopsies that could explain the lack of dermonecrosis due to *N. nigricollis* venom and the less severe skin tissue damage seen in biopsies injected with the other venoms. 

As a venom control for this study, we used venom from the non-necrotic cobra *N. haje*, which, as expected, did not demonstrate any evidence of macroscopic damage to skin models, although epidermal changes were observed histologically at 48 h. Pyknotic nuclei were seen deep in the dermis in the *N. haje* venom-treated biopsy at 168 h and there was evidence of apoptosis from anti-active caspase-3 fluorescence immunostaining and TUNEL. These results indicate cellular damage is still occurring from non-necrotic venoms, and these findings may reflect the presence of cytotoxic three-finger toxins in the venom of *N. haje* [50,51,52]. 

Neutralization of venom-induced dermonecrosis by F(ab’)_2_ or IgG antivenom has been investigated in a mouse model with intravenous administration of antivenom immediately following intradermal injection of *N. nigricollis* venom [49]. These interventions were found to neutralise dermonecrosis only partially, and only when IgG antivenom was administered [49]. In this human skin study, we performed a similar experiment, in which F(ab’)_2_ SAIMR monovalent or polyvalent antivenom was injected subcutaneously one hour following subcutaneous injections of the human skin biopsies with *E. ocellatus* or *N. nigricollis* venoms. While the histological analyses failed to identify evidence that the antivenoms had reversed venom-induced damage, we observed a notable down-regulation of transcription and secretion of many key chemokines, cytokines, and growth factors only in the antivenom-treated envenomed biopsies. 

Results, mostly from murine studies of Latin American snake venoms, strongly suggest that inflammatory infiltrates in the hypodermis and basal dermis contribute to the pathogenesis of snakebite dermonecrosis [21,22,24,33,49], and that inflammatory mediators may contribute to local pathology [30,32,53]. Given the significant differences between mouse and human immune system development, activation, and response [54], we used RNA-seq and the Luminex™ Human Cytokine Magnetic 30-Plex Panel to investigate both gene expression and the secretion of chemokines, cytokines, and growth factors from human skin biopsies that had been subcutaneously injected with venom (*E. ocellatus*, *B. arietans*, *N. nigricollis*, or *N. haje*). 

Chemokines are small polypeptides (8–10 kDa) synthesized by several cell types, including keratinocytes and fibroblasts in the skin, and they function mainly as chemoattractants for phagocytic cells, recruiting leucocytes from the blood to sites of injury, as has been observed from inflammation induced by intraperitoneal injections of *B. asper* or *B. jararaca* venom in mouse models [28,55]. C-X-C chemokines with a glutamic acid-leucine-arginine motif preferentially attract neutrophils, and possibly lymphocytes [56], whereas those without this motif preferentially attract only lymphocytes. From the RNA-seq results for venom-injected skin biopsies, C-X-C chemokine CXCL5 was upregulated to the greatest extent in all venom treatments in comparison to all other pro-inflammatory genes. CXCL5 activates CXC receptor 2 (CXCR2)-bearing cells, which include neutrophils, recruiting these cells to sites of inflammation, in addition to promoting connective tissue remodelling [57]. CXCL5 also contributes to skin hypersensitivity and pain [58]. Therefore, CXCL5 could be a key regulator in the recruitment of neutrophils in response to venom inflammation and could be a mediator of venom-induced local pain or hyperalgesia in humans. 

From the Human Cytokine Magnetic 30-Plex Panel results using biopsy media, the C-C chemokines MIP1-ALPHA, RANTES, MCP-1, and the C-X-C chemokine MIG exhibited peak levels at or beyond 24 h following venom injections. MCP-1 levels were elevated after *B. arietans* envenoming in a peritonitis mouse model, and *Bothrops pirajai* venom SVMPs have been reported to increase MCP-1 and MIG production in the human whole-blood model [59], the former reflecting macrophage migration [60]. MCP-1 serves as a macrophage chemoattractant and activator in human skin injuries, triggering inflammatory and pro-growth cytokine production. Lymphocyte migration has also been correlated with MCP-1, MIG, and IP-10 expression during wound healing after 96 h [61]. 

Inflammation-related immune response and tissue repair are underpinned by leukocyte-produced anti- and pro-inflammatory cytokines, which are relatively low-molecular weight proteins (8–25 kDa) that regulate the amplitude and duration of immune response [62]. We observed heightened levels of pro-inflammatory cytokines IL-1β, IL-1RA, IL-12, G-CSF/CSF-3, and GM-CSF detected in the media of the venom-injected human skin biopsies. 

Following *E. ocellatus* and *B. arietans* venom injections in biopsies, IL-1β levels increased within 48 to 72 h, and a negligible increase was noted 24 h after *N. nigricollis* and *N. haje* venom injections, but this response dropped within 24 h after peaking for all venoms. The cytokine response will be reduced in this model compared to in vivo, as further inflammatory and immune-modulatory cells cannot be recruited from the circulation to magnify the reaction. IL-1β functions include prostaglandin synthesis and chemokine discharge regulation, lymphocyte activation, macrophage stimulation, and promotion of leukocyte adhesion to endothelial cells [21,63]. IL-1β can activate hyperalgesia, peripheral oedema enlargement, and allodynia [64,65]. Intramuscular injection of *B. asper* venom also induced IL-1β in mice, however the use of anti-cytokine antibodies found that IL-1β likely does not play a significant role in dermonecrosis pathogenesis from *B. asper* venom, although it could be involved in reparative and regenerative responses to tissue damage from venom [66]. 

The elevated G-CSF and GM-CSF could also be promoting tissue repair after envenoming [67], and we found elevated levels of G-CSF and GM-CSF within 48 h following *N. nigricollis* and *N. haje* venom injections. Immune response activation and pro-inflammatory cytokine release triggers GM-CSF production by macrophages, mast and endothelial cells, T cells, and fibroblasts [68]. Its discharge into skin by keratinocytes is induced by injuries, tumour stimulation, or inflammatory skin conditions, including promoting keratinocyte proliferation and skin repair in lepromatous leprosy, according to in vitro research by Kaplan et al. (1992) [69]. GM-CSF was found to be elevated in mice treated with the basic PLA_2_ from the venom of the eastern diamondback rattlesnake, *Crotalus adamanteus*, with a suggested role in skin wound healing [70]. 

We detected downregulation in gene expression of IL-6 in human skin biopsies injected with venoms. In contrast, IL-6 was identified in the inflammatory exudates of mice intramuscularly injected with *B. asper* venom and toxins [21,53,66]. Similarly, increased levels of IL-6 have been identified from intraperitoneal injections of *B. arietans* venom in a mouse model [71]. IL-6 was also detected in blister exudate collected from human victims of *B. atrox* envenoming [72]. The IL-6 inflammatory response could be originating from cells absent in our model but which would be present in a whole organism or more complex model with multiple tissue types and vascularisation. Regrettably, we could not investigate IL-6 expression in our biopsy media due to a lack of IL-6 specificity in the Luminex™ Human Cytokine Magnetic 30-Plex Panel. 

Snake venom, especially viperid venoms, contains SVMPs that cleave basement membrane components, degrading the ECM, while inflammation-related ECM component proteolysis generates growth factors (e.g., IGF, VEGF, FGF, TGF-β, HGF, and PDGF) that promote cell activation, differentiation, and proliferation [73,74]. VEGFs are key regulators of angiogenesis and might also be contributing to the tissue repair processes, as well as increasing vascular permeability [75]. The inflammatory exudate produced from envenoming, especially in viperid snakebite, is known to increase vascular permeability [30], and could mediate the spread of toxins. VEGF-A gene expression was upregulated throughout the large majority of *E. ocellatus* and *B. arietans* venom-injected biopsy time points (9/10), and at 48 h following the injection of *N. nigricollis* venom. Secreted VEGF-A in the cell media of necrotic venom-injected biopsies (*E. ocellatus*, *B. arietans*, and *N. nigricollis*) peaked at 96 h, before rapidly decreasing to normal levels. 

Our RNA-seq approach allowed for the identification of other biological processes, in addition to inflammation, that are triggered during envenoming. We noted upregulation of biological processes associated with keratinocyte differentiation and tissue development from the venom-injected human biopsy transcriptome, including upregulation of gene *KRT23* for all necrotic venom treatments. KRT23 mediates epithelial–mesenchymal transition and proliferation of cells [76], which is suggestive of hyperplasia associated with wound healing. 

We detected upregulation of multiple S100 genes in the venom-injected human skin biopsies. Elevated levels of S100 proteins have been found in exudate collected from viperid *B. asper* envenoming in mouse models [30] and from blister exudate collected from *B. atrox*-envenomed human snakebite victims [77]. S100 proteins are released from the cytoplasm during cell damage and stress, interacting with RAGE (Receptor for Advanced Glycation End products) to induce the production of pro-inflammatory cytokines, leading to the migration of neutrophils, monocytes, and macrophages [78]. The S100A8-S100A9 complex mediates complement factor C3 and has been found to be selectively upregulated in lesional psoriatic skin disease [79]. 

We also detected upregulated expression of TIMP1, a matrix- and disintegrin-metalloproteinase inhibitor, in all necrotic venom-injected biopsies, which could be in response to cell recognition of direct damage by SVMPs, or concomitantly released with endogenous matrix metalloproteinases [21]. This RNA-seq dataset provides a comprehensive list of endogenous genes that could be examined and validated in future studies to determine their contributions to venom-induced dermonecrosis in human tissues.

The tested model occupies the research space between clinical observations and in vivo animal models. An obvious weakness of this model is the inability to mimic intravenous interventions, which, in the field of snakebite therapy, places clear limits on model utility. Another limitation is that the need for experimentation within a day of biopsy excision, coupled with the costs of acquiring human skin biopsies, will inevitably restrict experimentation to well-funded groups close to surgical facilities. The lack of vascularisation of the ex vivo biopsies might also exert physiological conditions restricting the duration of experiments and the cells/systems that can be validly examined, although our comparisons between control and venom-injected biopsies in terms of skin morphology, cell death, and skin-cell expression of a wide variety of cytokines, chemokines, and growth factors identified that the biopsies were physiologically viable and responsive to tissue injury for at least six days after excision. The ability to capture these datasets from human skin tissue therefore represents a significant and clinically relevant advancement over existing murine models of local envenoming. Nevertheless, it is important to consider the lack of vascularisation and our limited sample size in the interpretations of the results gained using the model presented here.

## 4. Conclusions

Our ex vivo human skin model shows promise as a valuable tool for studying the progression and treatment of cytotoxic snakebite envenoming. From observable morphologic damage and quantification of pro-inflammatory mediators, our ex vivo skin model confirms cytotoxicity following *E. ocellatus*, *B. arietans*, and *N. nigricollis* envenoming, while showing that, although there is no observed clinical dermonecrosis from *N. haje* venom in comparison to the other three venoms, the venom is still responsible for causing apoptosis and inflammation. Compared to murine models of local envenomation, ex vivo human biopsies demonstrated similar histological signs of epidermal necrosis and apoptosis and were stable over long time periods (up to 168 h). Once standardized, this model could be used to assess the effectiveness of antivenoms and other complementary therapies against venom-induced dermonecrosis. Future studies could valuably explore the utility of this ex vivo human skin model to identify venom toxins that cause, and testing medicinal candidates to prevent, the onset of venom-induced pain, oedema, swelling, blistering, and other pathological processes associated with the early effects of local snakebite envenoming.

## 5. Material and Methods

### 5.1. Snake Venom

Snake venom was sourced from animals maintained in the herpetarium of the Liverpool School of Tropical Medicine (LSTM). This facility and its snake husbandry protocols are approved and inspected by the UK Home Office (establishment licence number 40/9074 [X2OA6D134]) and LSTM’s Animal Welfare and Ethical Review Body. Venom was extracted from four African snake species (Table 1), three species known to cause local tissue necrosis, namely, the vipers *Echis ocellatus* and *Bitis arietans* and the spitting cobra *Naja nigricollis*, and one cobra species, *N. haje*, included as a control because its venom does not cause dermal necrosis. (*N. haje* venom is neurotoxic.) All venoms were pools from multiple individuals; *E. ocellatus* venom was pooled from at least 49 specimens, *B. arietans* venom was pooled from 10–12 specimens, *N. haje* venom was pooled from three specimens, and *N. nigricollis* venom was pooled from five specimens. All pools were from snakes of adult size and included both sexes. Venoms were lyophilised and then resuspended in phosphate buffered saline (PBS) for downstream experiments.

### 5.2. Human Skin Collection

Biopsies of human skin samples were provided by Genoskin (Toulouse, France). Genoskin collected anonymised human skin samples from two female donors, aged 40 and 47 years old, each of whom had undergone an abdominoplasty procedure and had provided written informed consent. Donors did not have any record of allergies or dermatological disorders and had not used glucocorticoid or steroid treatments. Full ethical approval was obtained from the French ethical research committee (Comite de Protection des Personnes) (AC-2011-1443), and authorisation was also provided by the French Ministry of Research. All studies were conducted according to the Declaration of Helsinki protocols and in compliance with the UK Human Tissue Act (LSTM research tissue bank, REC ref. 11/H1002/9). The biopsies were prepared by Genoskin and delivered (under refrigeration) to Liverpool the day after surgical collection. Experiments were initiated immediately upon biopsy arrival. The HypoSkin^®^ model, developed as an ex vivo human skin model by Genoskin (France), comprises biopsies 15 mm in diameter and 10 mm thick, consisting of three skin layers (epidermis, dermis, and subcutis) and subcutaneous fat tissue. 

### 5.3. Skin Culture Conditions and Treatments

Immediately upon arrival, the human skin biopsies for both venom and antivenom intervention experiments (*n* = 84) were cultured under standard mammalian cell conditions and held at the air–liquid interface (37 °C, 5% CO_2_, and maximum humidity). Two biopsies were immediately fixed in 10% buffered formalin for 48 h at room temperature and routinely processed for formalin-fixed, paraffin-embedding (FFPE) (these are referred to as the uncultured controls). In addition, two biopsies were left untreated as negative controls over the 168 h (7-day) time course. Three separate venom experiments were conducted with the biopsies, a venom-only time course from 0 h to 168 h, a venom-only time course from 0 h to 48 h, and an antivenom intervention from 0 to 48 h (Appendix A).

Venom doses, based on three times the Minimum Necrotising Dose (previously determined in mouse models of envenoming [80]) (Table 1), were subcutaneously injected into skin biopsies (*n* = 2 for each venom) in a volume of 20 µL using a Hamilton LT 250 μL syringe and 27G needle. A venom dose of 156 µg was chosen for *N. haje*, as no MND data were available. At ten time points following venom subcutaneous injections (6, 12, 24, 36, 48, 72, 96, 120, 144, and 168 h), one skin biopsy was FFPE-processed for histological analysis and a second biopsy taken and stored in RNAlater (Sigma, Lot; MKCG5161, St. Louis, MI, USA) for RNA sequencing.

To investigate the efficacy of antivenom in this model, biopsies were first injected subcutaneously with *E. ocellatus* venom (120 µg) or *N. nigricollis* venom (165 µg), then one hour later, the SAIMR Echis monovalent (Lot, BC00147, Exp JAN 2016; South African Vaccine Producers; vial concentration 52 mg/mL) or SAIMR polyvalent (Lot, BF0546, Exp Nov 2017 South African Vaccine Producers; vial concentration 103 mg/mL) antivenom was subcutaneously injected (20 µL of neat antivenom/biopsy) into the *E. ocellatus* or *N. nigricollis* ‘envenomed’ biopsy, respectively. Antivenom controls consisted of Equine IgG (BioRad, Hercules, CA, USA), and the SAIMR Echis and SAIMR Polyvalent antivenoms. Forty-eight hours following antivenom and control administrations, one skin biopsy was FFPE-processed for histological analysis and a second stored in RNAlater (Sigma, Lot; MKCG5161) for RNA sequencing.

### 5.4. Haematoxylin and Eosin Staining and Imaging

Histological examination of venom-induced dermonecrosis over the 168 h time course and after antivenom intervention was conducted on the FFPE skin biopsies. Histological sections from the 168 h venom-induced dermonecrosis were processed (sectioned at 5 μm using a microtome (Leica, Wetzlar, Germany) and stained with haematoxylin and eosin (H/E), with images taken with an optical microscope (Leica DMi1), and analysed by Genoskin. Histological sections from the 48 h venom and antivenom intervention were processed at the University of Liverpool School of Veterinary Science, where 4 µm paraffin sections were mounted on glass slides and dewaxed in xylene for 5 min before rehydration through a series of ethanol solutions (96%, 85%, and 70%) to distilled water. Sections were stained for 5 min in haematoxylin (HD1475, TCS Biosciences Ltd., Buckingham, UK) before ‘blueing’ in tap water for 6 min and staining in eosin (TCS Biosciences Ltd., HS250-1L) for 2 min. Excess eosin was removed using 96% ethanol and then the sections were dehydrated in 100% ethanol, followed by xylene. Slides were cover-slipped using DPX mountant (CellPath, phthalate free SEA-1304-00A); Newton, UK). Sections were then analysed by a veterinary pathologist (G.L). The skin biopsies were bisected at the end of the experiment, and one of the halves placed in formalin for histological examination. The cut surface was at the centre of the biopsy (site of inoculation) and complete transverse sections of this orientation were examined, comparing changes seen at different time points to those seen in control experiments, by use of an optical microscope (Olympus, BX43, Tokyo, Japan)

### 5.5. Anti-Active Caspase-3 Fluorescence Immunostaining

Skin biopsies were stained using immunofluorescence to identify the presence of active apoptotic caspase-3, suggestive of cell apoptosis, 168 h following venom subcutaneous injections. Five micrometre sections of FFPE skin biopsies were incubated in blocking buffer (Goat serum, ab7481, Abcam, Cambridge, UK) for 40 min at 37 °C, and thereafter, sections were washed with PBS for five minutes, a total of three times, before an overnight incubation at 4 °C with a cleaved caspase-3 antibody (Rabbit polyclonal; Abcam ab2302) at a dilution of 1:100 in blocking buffer with gentle agitation. Sections were washed again as previously described and incubated with an HRP-conjugated secondary antibody at 1:500 for one hour at room temperature (RT). Following another wash, fluoroshield mounting medium with DAPI (Abcam, 104139) was applied. Biopsy sections, both with and without venom treatments, were examined via microscopy (Leica DMi1, Wetzlar, Germany) and processed by ImageJ software version 1.54.

### 5.6. TUNEL Assay

TUNEL (Terminal deoxynucleotidyl transferase dUTP Nick End Labelling) was performed as a complementary procedure to active apoptotic caspase-3 staining for the detection of apoptotic or necrotic cells present in skin biopsies 168 h following venom subcutaneous injections. This was performed using the ApopTag^®^ Fluorescein In Situ Apoptosis Detection Kit (Ref. S7110, Millipore, Burlington, MA, USA), and following the manufacturer’s protocol. Briefly, sections of FFPE skin biopsies 5 μm thick were washed with xylene for five minutes (three changes), absolute ethanol for five minutes (two changes), 95% ethanol (once), 70% ethanol for three minutes (once), and then received a final wash with PBS for five minutes (one change). Digesting enzyme proteinase K (20 μg/mL) was applied to the pre-treated tissues for 15 min at RT in a Coplin jar, followed by two washes with PBS for two minutes, and excess liquid was gently tapped off. An ApopTag^®^ equilibration buffer (Ref, S7106) was then immediately applied directly to tissues and incubated for ten seconds at RT. Excess liquid was gently tapped off again and terminal deoxynucleotide transferase ApopTag^®^ (TdT) enzyme (Ref, 90418) pipetted onto the sections and incubated for one hour at 37 °C. Sections were transferred to a Coplin jar containing an ApopTag^®^ stop/wash buffer (Ref, S7108), with gentle agitation for 15 s, and then incubated at RT for ten minutes. After incubation, sections were washed three times with PBS for one minute per wash, and warmed anti-digoxigenin conjugate was applied to the slide, which was then incubated for 30 min at RT in a humidified chamber while avoiding exposure to the light. Sections were then washed with PBS for two minutes at RT (four changes), before the application of a mounting medium containing 0.5–1.0 μg/mL DAPI. Biopsy sections, both with and without venom treatments, were examined with a Leica DM5000 microscope and captured images processed with ImageJ software version 1.54. 

### 5.7. Quantification of Fluorescence Intensities for Skin Biopsies

Fluorescence intensities were measured using ImageJ software version 1.54 (Java-based image processing) for all skin biopsies stained with DAPI and TUNEL. Measurements were applied to the entire image, with the area integrated intensity and mean grey value determined for both areas with fluorescence and areas without fluorescence (as background). Corrected total cell florescence (CTCF) was calculated as CTCF = Integrated Density—(Area of selected cell × Mean fluorescence of background reading) and plotted with GraphPad. For all analyses, the significance was assessed by analysis of variance (ANOVA), followed by Dunnett’s multiple comparison test. A *p* ≤ 0.05 was considered significant for all statistical tests. All statistical analyses were conducted using GraphPad Prism 9.

### 5.8. RNA Sequencing of Skin Biopsies and Bioinformatic Analyses

To evaluate changes in the expression of pro-inflammatory genes in venom-treated skin biopsies after 6, 12, 24, 36, and 48 h, in comparison to the untreated control, RNA sequencing (RNA-seq) was performed. Total RNA was extracted from the biopsies and RNA quality evaluated on an Agilent Bioanalyzer (RIN required to be >7 for library preparation). RNA-seq library preparation and sequencing were performed commercially by GENEWIZ (Azenta Life Sciences, Leipzig, Germany). Briefly, ribosomal RNA was depleted, the remaining RNA fragmented, and random priming used for first strand cDNA synthesis. This was followed by second strand cDNA synthesis, end repair, 5′ phosphorylation, and dA-tailing. Illumina adaptors were ligated to cDNA fragments and fragments enriched by PCR before being multiplexed and sequenced together on a HiSeq 2500 instrument single lane to obtain 150 bp paired-end reads. The sequencing data have been submitted to NCBI under BioProject ID PRJNA855360 and BioSample SAMN29490060.

Low-quality reads and contaminant adaptors were removed with Trimmomatic v.0.36 [81], and a total of 33–56 million high-quality reads per sample were aligned to the *Homo sapiens* GRCh38 reference genome available on ENSEMBL, using the STAR aligner v2.5.2b [82]. Unique gene hit counts were calculated using the feature ‘Counts’ from the Subread package v1.5.2. These gene counts were then used as input into GFOLD [83] to determine the fold-change between genes in untreated and venom-treated conditions. Log2 expression values visualized from heatmaps were created with GraphPad Prism 9. The top 50 upregulated genes for each venom at each time point were compared with Venny 2.1 [84]. Genes upregulated at least 2-fold were evaluated with the Gene Functional Annotation Tool from the DAVID Bioinformatics Database v2021 [85], gene ontology option GOTERM_BP_ALL was selected, and the first eight terms with a *p*-value < 0.05 were chosen. Biological process (BP) gene ontology terms related to skin were visualized in a GraphPad Prism 9 heatmap. 

### 5.9. Quantification of Human Pro-Inflammatory Markers in Biopsy Culture Medium by Multiplex Bead Array

Complementary to RNA-seq experiments, the Human Cytokine Magnetic 30-Plex Panel (based on xMAP™ technology) (Cat, LHC6003M, ThermoFisher, Vienna, Austria) was used to quantify pro-inflammatory markers (chemokines, cytokines, and growth factors) in culture media collected 6, 12, 24, 36, 48, 72, 96, 120, 144, and 168 h following venom treatments. The panel enabled quantification of EGF, Eotaxin, FGF basic, G-CSF, GM-CSF, HGF, IFN-α, IFN-γ, IL-1RA, IL-1β, IL-2, IL-2r, IL-4, IL-5, IL-6, IL-7, IL-8, IL-10, IL-12 (p40/p70), IL-13, IL-15, IL-17, IP-10, MCP-1, MIG, MIP-1α, MIP-1β, RANTES, TNF-α, and VEGF. Standards and samples were prepared and reconstituted following the manufacturer’s instructions. Briefly, antibody beads were added to each well and incubated on the provided magnet for 60 s before the liquid was decanted and washed twice with 200 μL of washing buffer. Next, 50 μL of incubation buffer was added, followed by 100 μL of diluted standard and blank (PBS) in duplicate for all samples. Then, 50 μL of assay diluent was added before the addition of 50 μL of neat culture medium samples. The plate was covered and kept on a rocker overnight at 4 °C. The following day, the plate was washed as described and 100 μL of biotinylated detector antibody added and incubated for one hour at RT, followed by three washes and the addition of 150 μL of wash buffer. Luminex^®^ 100/200 technology (Mag-Plex^®^-Avidin Microspheres) was used to measure the resulting bead fluorescence. xPonent Luminex 4.0 software was used to analyse results, and the mean and standard error were calculated for each group. The *p*-value of the reading of pro-inflammatory markers of biopsies treated with snake venoms at 48 h and 120 h were calculated and compared with that of the untreated control. Significant differences were assessed by analysis of variance (ANOVA), followed by Dunnett’s multiple comparison test. A *p* ≤ 0.05 was considered significant for all statistical tests. All statistical analyses were conducted using GraphPad Prism 9.

## Figures and Tables

**Figure 1 toxins-16-00276-f001:**
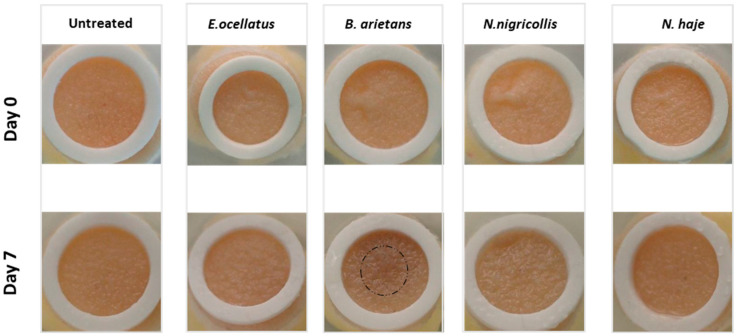
Macroscopic changes in venom-injected biopsies. Venoms (*E. ocellatus* 120 μg; *B. arietans* 195 μg; *N. nigricollis* 165 μg; *N. haje* 156 μg) were injected into Hyposkin^®^ biopsies 15 mm in diameter and 10 mm thick, consisting of three skin layers (epidermis, dermis, and subcutis), and observed on day seven (168 h) for macroscopic changes. Day 0 refers to the time before subcutaneous injection. After seven days in culture, the untreated control and the biopsy injected with *N. haje* venom did not display any morphological changes. All other biopsies exhibited lesions, the most pronounced being caused by *B. arietans* venom (black circle shows area of change). Photographs captured with an MC170 HD camera.

**Figure 2 toxins-16-00276-f002:**
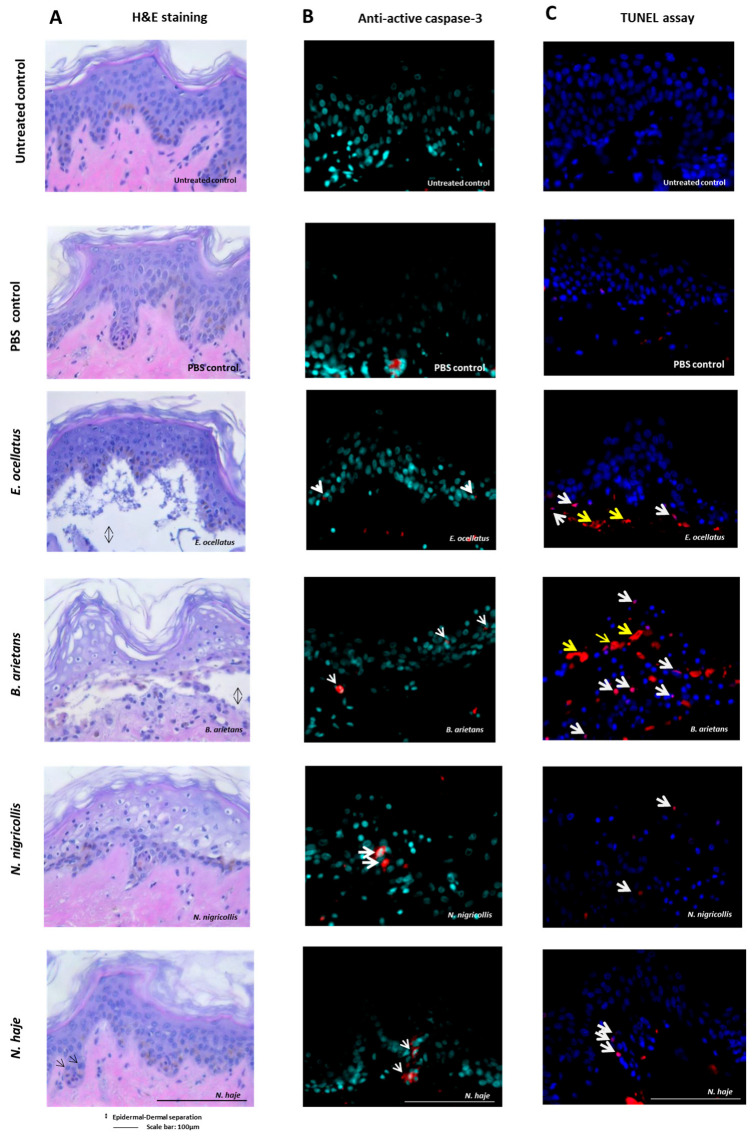
Structural integrity and viability assessments of HypoSkin^®^ models seven days following the subcutaneous injection of snake venoms. HypoSkin^®^ models at day seven following subcutaneous injection of snake venoms (*E. ocellatus* 120 μg; *B. arietans* 195 μg; *N. nigricollis* 165 μg; *N. haje* 156 μg): (**A**) haematoxylin and eosin staining, (**B**) anti-active caspase-3 fluorescence immunostaining, and (**C**) DNA fragmentation determination by TUNEL staining. Black arrows indicate pyknotic nuclei in epidermal crests for the *N. haje* venom-injected biopsy. The examination of apoptotic cells using anti-active caspase-3 fluorescence immunostaining detected apoptotic cells (white arrows) in all conditions except the untreated control and the PBS-injected control. TUNEL staining confirmed the presence of either apoptotic cells (white arrows) or necrotic cells (yellow arrows) in all biopsies injected with venom. Untreated biopsies did not present any apoptotic or necrotic cells. Images were taken at 40× magnification; this showed the overall extent of the epidermal loss (erosion/ulceration).

**Figure 3 toxins-16-00276-f003:**
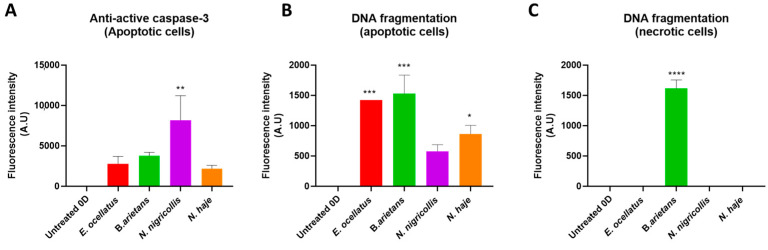
Measurements of the fluorescence intensities of HypoSkin^®^ models seven days following the subcutaneous injection of snake venoms. HypoSkin^®^ models at day seven following subcutaneous injection of snake venoms (*E. ocellatus* 120 μg; *B. arietans* 195 μg; *N. nigricollis* 165 μg; *N. haje* 156 μg), showing: (**A**) anti-active caspase-3 fluorescence immunostaining, and (**B**,**C**) DNA fragmentation determination by TUNEL staining. ImageJ software version 1.54 was used to quantify the fluorescence of apoptotic cells following anti-active caspase-3 fluorescence immunostaining, and the presence of apoptotic cells or necrotic cells following TUNEL staining. Bars represent means of triplicate measurements, and error bars represent standard deviation. Statistical significance was determined via ANOVA with Dunnett’s multiple comparison test, with asterisks indicating thresholds of * *p* ≤ 0.01, ** *p* < 0.008, *** *p* < 0.0002, and **** *p* < 0.0001. Note the different *y*-axis scales in (**A**) compared with (**B**,**C**).

**Figure 4 toxins-16-00276-f004:**
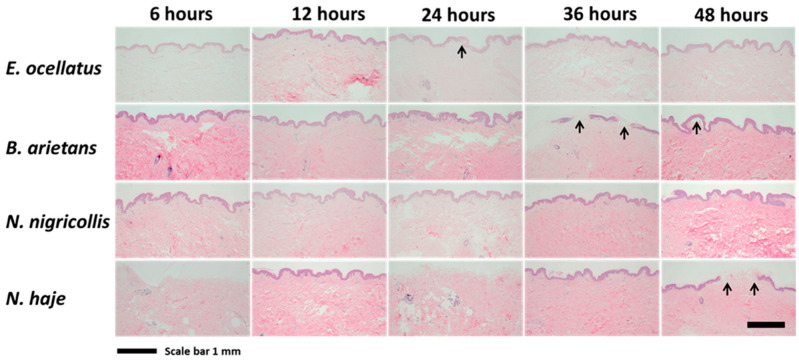
Histology of biopsies at time points 6, 12, 24, 36, and 48 h following subcutaneous injection of snake venoms. After venoms (*E. ocellatus* 120 μg; *B. arietans* 195 μg; *N. nigricollis* 165 μg; *N. haje* 156 μg) were subcutaneously injected into HypoSkin^®^ models, 5 μm skin sections were stained with haematoxylin and eosin; these are shown at 4× magnification. Variation is seen between samples, even those treated with the same venom, and later time points do not always show a progression of epidermis degeneration. Images were taken at 4× magnification to demonstrate the overall extent of epidermal loss (erosion/ulceration).

**Figure 5 toxins-16-00276-f005:**
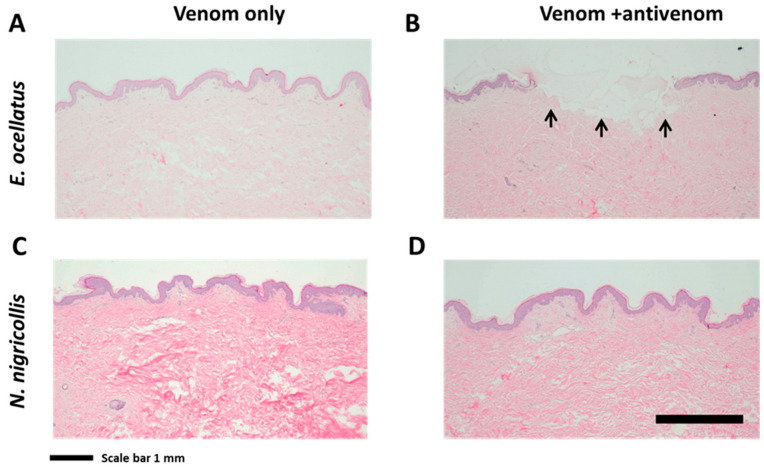
Histologies of venom- and antivenom-injected biopsies. Images show HypoSkin^®^ sections 48 h following subcutaneous injection of snake venom only, or venom then followed by antivenom at 48 h: (**A**) *E. ocellatus*, venom only (120 μg); (**B**) *E. ocellatus*, venom (120 μg) followed by SAIMR Echis monovalent antivenom (20 µL) at 48 h; (**C**) *N. nigricollis*, venom only (165 μg); and (**D**) *N. nigricollis*, venom (165 μg) followed by SAIMR polyvalent antivenom at 48 h. Skin sections were stained with haematoxylin and eosin, and images taken at 4× magnification. Arrows indicate locations of epidermal loss (ulceration), which was present even with antivenom treatment at 48 h. Images were taken at 4× magnification to demonstrate the overall extent of the epidermal loss (erosion/ulceration).

**Figure 6 toxins-16-00276-f006:**
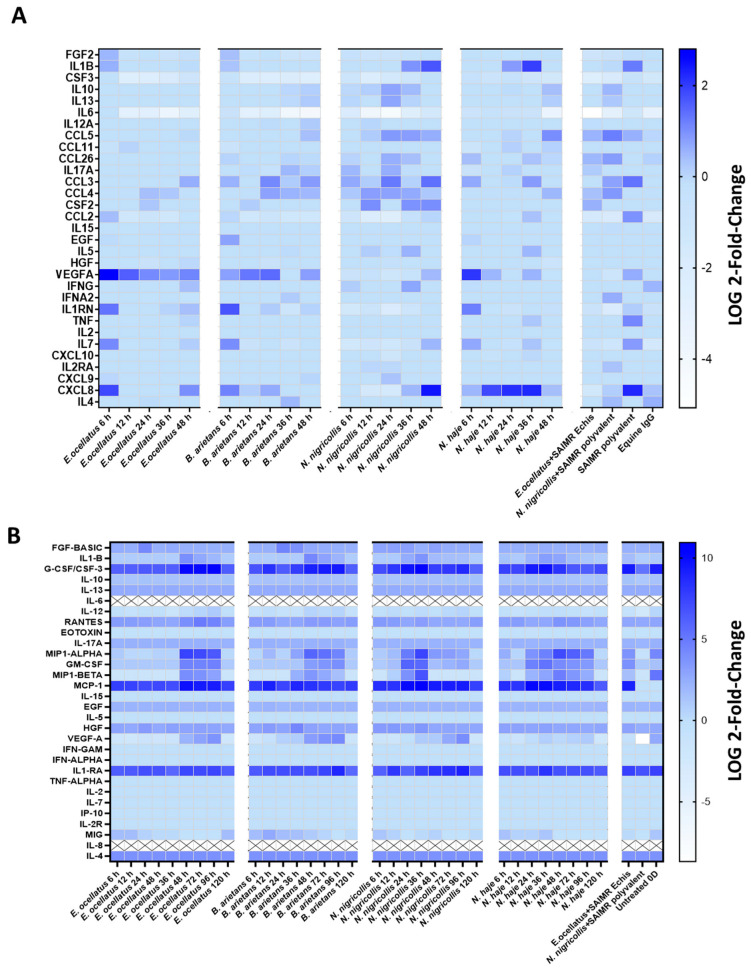
Changes in pro-inflammatory chemokines, cytokines, and growth factors following *Echis ocellatus*, *Bitis arietans*, *Naja nigricollis*, and *N. haje* venom subcutaneous injections in HypoSkin^®^ biopsies. Pro-inflammatory responses were quantified by (**A**) RNA sequencing of HypoSkin^®^ tissue following venom and antivenom treatments over 48 h, and by using the (**B**) Human Cytokine Magnetic 30-Plex Panel, with culture media collected at 6, 12, 24, 36, 48, 72, 96, and 120 h following venom treatments.

**Figure 7 toxins-16-00276-f007:**
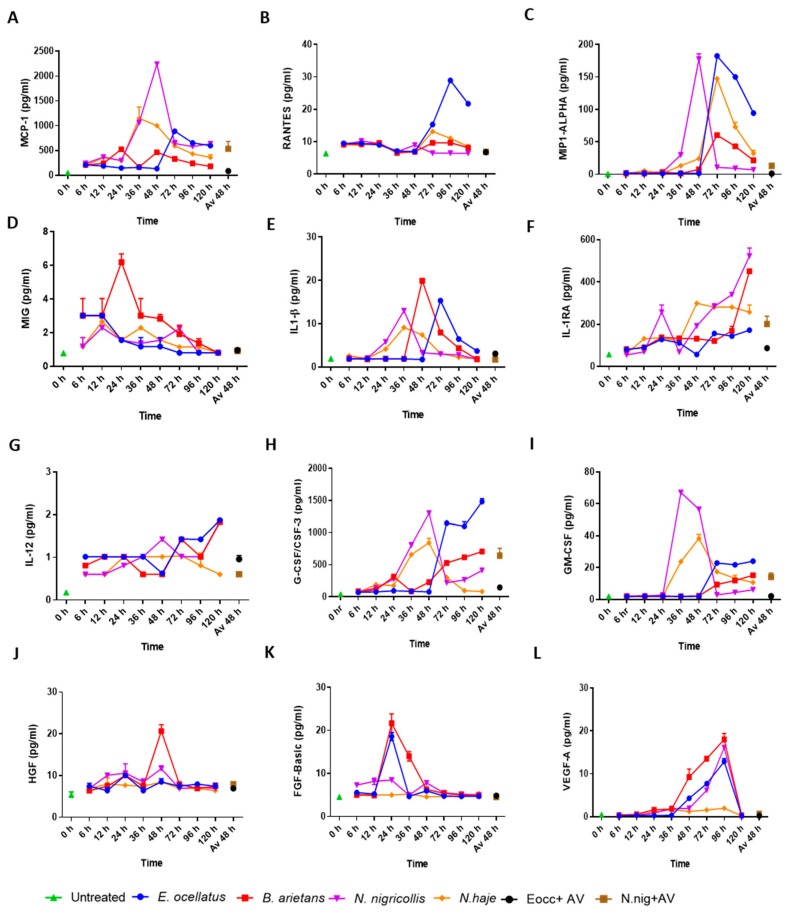
Dynamic changes in the expression, and secretion into culture media, of pro-inflammatory chemokines, cytokines, and growth factors of human skin biopsies injected with viper (*Echis ocellatus* and *Bitis arietans*) and elapid (*Naja nigricollis* and *N. haje*) venoms. The Human Cytokine Magnetic 30-Plex Panel for the Luminex™ platform was used to quantify pro-inflammatory responses in culture media collected from the venom-injected HypoSkin^®^ biopsies over 120 h and 48 h following antivenom intervention. Data are shown for four chemokines, namely, (**A**) MCP-1, (**B**) RANTES, (**C**) MIP1-ALPHA, and (**D**) MIG; five cytokines, namely, (**E**) IL-1β, (**F**) IL-1RA, (**G**) IL-12, (**H**) G-CSF/CSF-3, and (**I**) GM-CSF; and three growth factors, namely, (**J**) HGF, (**K**) FGF-Basic, and (**L**) VEGF-A. Cell-culture media collected from untreated HypoSkin^®^ biopsies were used for response comparisons. Eocc + AV, *E. ocellatus* + SAIMR Echis antivenom; N.nig + AV, *N. nigricollis* + SAIMR polyvalent antivenom.

**Table 1 toxins-16-00276-t001:** African snake species and venom doses.

Snake Species	Family	Common Name	Origin	Venom Dose 3× (MND)
*E. ocellatus*	Viperidae	West African carpet viper	Nigeria	120 µg
*B. arietans*	Viperidae	African puff adder	Nigeria	195 µg
*N. nigricollis*	Elapidae	Black-necked spitting cobra	Nigeria	165 µg
*N. haje*	Elapidae	Egyptian cobra	Uganda	ND

MND refers to the minimum amount of venom required to produce a 5 mm diameter necrotic lesion in mouse skin 72 h after intradermal injection of venom.

## Data Availability

The data presented in this study are available in this article and Appendix A.

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
