# Peer review of "Investigating Snake-Venom-Induced Dermonecrosis and Inflammation Using an Ex Vivo Human Skin Model"

_toxins, 2024, doi:10.3390/toxins16060276_

Round 1
Reviewer 1 Report
Comments and Suggestions for Authors
Thank you for sending me a very interesting manuscript for review. Testing the effects of snake venom is a highly debated and still relevant topic. The issue of using mice and other animal models is complex from many perspectives, whether ethical or comparability of results between animals and humans. That is why I was interested in your study, which offers an alternative for future research. I appreciate the very well performed experiments using modern laboratory techniques. Nevertheless, I have a few comments on your manuscript, but of a rather formal character.
Throughout the text, there are often different fonts, even within the same sentence, different line spacing or unnecessary spaces between words. Please pay attention to the formatting of the text, it unnecessarily reduces the quality of a great article.
The methodology of sample preparation and in particular the subsequent tests on injected skin are described in great detail. The only thing I would recommend is a simple diagram of the whole experiment. It is not entirely clear from the description of the procedure how many samples were injected with venom, how many were subsequently treated with antivenom and when which samples were monitored in which test. All the information is in the text, but it is a bit difficult to track down the information for a general overview. It is also not stated if the venom was taken from just one individual of each species or if there were multiple snakes or if the animals were of comparable age/size.
In the results section, Figure 1 with photographs of skin changes is presented. Unfortunately, the mobile phone photos are not of very high quality and the changes shown in the caption are difficult to detect in the poor quality photos, except perhaps for the sample for Bitis arietans. Consider using these photos in this article and for future publications using a camera with the same lighting for all photographed specimens.
The discussion contains some paragraphs that by their character would belong more in an introduction. For example, a paragraph on what chemokines are. I understand that you are trying to provide the reader with relevant information to follow up on your results, however, this information should have been earlier in the paper.
I would have appreciated a paragraph on the limitations of your chosen method at the end of the manuscript. Some of the limitations are mentioned in the discussion, and it is apparent that you are aware of them, but a summary paragraph would have been more appropriate.
Despite my comments I find your article very useful, the experiments are well designed and well done. I find your results innovative and hope you will continue to pursue this research.
Author Response
We thank the reviewer for their careful reading of the manuscript and their constructive remarks. Our response follows in italics to the reviewer’s comments in bold.
Thank you for sending me a very interesting manuscript for review. Testing the effects of snake venom is a highly debated and still relevant topic. The issue of using mice and other animal models is complex from many perspectives, whether ethical or comparability of results between animals and humans. That is why I was interested in your study, which offers an alternative for future research. I appreciate the very well performed experiments using modern laboratory techniques.
Author response:
Thank you very much for your kind comments.
Nevertheless, I have a few comments on your manuscript, but of a rather formal character.
Throughout the text, there are often different fonts, even within the same sentence, different line spacing or unnecessary spaces between words. Please pay attention to the formatting of the text, it unnecessarily reduces the quality of a great article.
Author response:
We thank the reviewer for this point and apologise for the formatting. We have now corrected this to improve the revised version of the manuscript.
The methodology of sample preparation and in particular the subsequent tests on injected skin are described in great detail. The only thing I would recommend is a simple diagram of the whole experiment. It is not entirely clear from the description of the procedure how many samples were injected with venom, how many were subsequently treated with antivenom and when which samples were monitored in which test. All the information is in the text, but it is a bit difficult to track down the information for a general overview.
We would like to thank the reviewer for this comment and suggestion. A general overview figure has now been included in the updated manuscript (Figure S1), and referenced in the Material and methods:
Line 153-156: Three separate venom experiments were conducted with the biopsies, a venom only time course from 0 hr to 168 hrs, a venom only time course from 0 hr to 48 hrs, and an antivenom intervention from 0 to 48 hrs (Figure S1).
It is also not stated if the venom was taken from just one individual of each species or if there were multiple snakes or if the animals were of comparable age/size.
In the results section, Figure 1 with photographs of skin changes is presented.
Echis ocellatus venom was pooled from multiple individuals, at least 49 specimens. Bitis arietans venom was pooled from 10-12 specimens. Naja haje venom was pooled from three specimens, and Naja nigricollis from five specimens. All snakes were of adult size and both sexes were represented. We have now added these details to the methods.
Line 126-129: All venoms were pools from multiple individuals, E. ocellatus venom was pooled from at least 49 specimens, B. arietans venom pooled from 10-12 specimens, N. haje venom pooled from three specimens, and N. nigricollis venom was pooled from five specimens. All pools were from snakes of adult size and included both sexes.
Unfortunately, the mobile phone photos are not of very high quality and the changes shown in the caption are difficult to detect in the poor quality photos, except perhaps for the sample for Bitis arietans. Consider using these photos in this article and for future publications using a camera with the same lighting for all photographed specimens.
We would like to thank the reviewer for this suggestion, but the photos were taken by professional cameras and the changes observed were very faint, only visible for one of the species. We have updated the photos and have added a circle and arrow to better indicate the visible venom-induced skin alterations.
The discussion contains some paragraphs that by their character would belong more in an introduction. For example, a paragraph on what chemokines are. I understand that you are trying to provide the reader with relevant information to follow up on your results, however, this information should have been earlier in the paper.
We thank the reviewer for this suggestion, although we prefer to retain the discussion in its current form. However, we have checked to made sure that the more general/background information we have written in the discussion relates to the context of envenoming. We feel this is important because of the scarcity of studies on this topic.
I would have appreciated a paragraph on the limitations of your chosen method at the end of the manuscript. Some of the limitations are mentioned in the discussion, and it is apparent that you are aware of them, but a summary paragraph would have been more appropriate.
We thank the reviewer for this suggestion, and have added a paragraph to the discussion to address the limitations of the ex vivo human skin model:
Line 718-732: The model occupies the research space between clinical observations and in vivo animal models. An obvious weakness of this model is the inability to mimic intravenous interventions, which, in the field of snakebite therapy, places clear limits on model utility. The need for experimentation within a day of biopsy excision, and the costs of human skin biopsies, will inevitably restrict experimentation to well-funded groups close to surgical facilities. The lack of vascularisation of the ex vivo biopsies might also exert physiological conditions restricting the duration of experiments and the cells/systems that can be validly examined, however, our comparisons between control and venom-injected biopsies in terms of skin morphology, cell death and skin-cell expression of a wide variety of cytokines, chemokines and growth factors identified that the biopsies were physiologically viable and responsive to tissue injury for at least six days after excision. The ability to capture these datasets from human skin tissue therefore represents a significant clinical relevance advancement over the murine models of local envenoming. Nevertheless, we have been careful to factor the lack of vascularisation into our interpretations of the results gained using this model, as well as our limited sample size.
Despite my comments I find your article very useful, the experiments are well designed and well done. I find your results innovative and hope you will continue to pursue this research.
Author response:
Thank you for your supportive comments.
Reviewer 2 Report
Comments and Suggestions for Authors
In the present report authors have used for the first time an ex-vivo non-perfused human skin model to broad investigate (both biochemically and morphologically) envenoming from both African vipers (Echis ocellatus and Bitis arietans) and cobras (Naja nigricollis and N. haje). The article was elegantly presented from introduction to discussion, and results suggested that the skin model applied was quite appropriate. These data were exciting as there are considerable ethical and financial concerns with similar assays that cause severe pain, harm and distress to experimental animal models. Therefore, this report not only successfully presents an original ex-vivo non-perfused human skin model, but it also solves compelling issue about the use of experimental animal models in research about human envenoming. Therefore, the present report provides convincing results showing how much does this model improves real pathophysiological responses from humans bitten by venomous animals.
Author Response
Author response:
Thank you for your supportive comments.
Reviewer 3 Report
Comments and Suggestions for Authors
Venom research is a crucial field, yet little innovation has been seen in assessing toxicity and developing new antivenom strategies. The gold standard methods are based on the use of animals or cells, which do not always mimic a real envenomation scenario. In this context, the present study brings a novel interesting and promising approach. The authors have used an ex-vivo human skin model to investigate snake venom-induced dermonecrosis. Interestingly, the study has focused on medically important African snakes. The manuscript is well-written and the central hypothesis is interesting with implications and potential to revolutionise the study of snake venoms, assessment of antivenoms and WHO guidelines. However, the study has limitations in the design of experimental, and statistical analysis and lack of quantitative data. Overall, most of the findings are based on qualitative data, which can be biased according to the section or area of the image described. Only figures 5 and 6 show quantitative data. However, they were not compared using statistical tests as well.
Although worth publishing, this study requires significant improvements from the authors.
1. Abstract section. Please specify the morphological changes and the clinical signs for better understanding.
2. Abstract section. What are the antivenom interventions?
3. Abstract section. reduce pro-inflammatory marker levels. All of them?
4. In the resubmission of the manuscript. please include number lines to facilitate feedback and peer review. process
5. Please include references supporting the following sentence: However, our current knowledge of local envenoming pathogenesis emanates primarily from subjecting murine models to venoms from Latin American snakes.
6. The introduction can be enriched with a parallel between the clinical pathology of viper and elapid snake venoms regarding local envenoming. What are the similarities and differences?
7. Please revise the following sentence: however, there are currently no alternate models to in vivo testing that accurately recapitulate local envenoming in snakebite victims. Despite the limitations, some studies use cell-based assays or other strategies that provide insights into local damage comparable to clinical manifestations observed in human victims. For example, the following researches (https://www.nature.com/articles/s41467-023-43510-w, https://www.ncbi.nlm.nih.gov/pmc/articles/PMC10850160/ ) have used epidermal keratinocyte, human cells to study myotoxicity, which is an important local manifestation caused by snake venoms. This point is important to discuss, but it worth mentioning different efforts to reduce the number of animals and develop approaches to assess venom toxicity and antivenom efficacy.
8. How was the number of biopsies studied per time point determined? How was the number of samples determined? In terms of statistical analyses, are two biopsies enough?
9. Did 84 biopsies include those used to study the efficacy of antivenom? This is not clear. To investigate the efficacy of antivenom in this model, biopsies were first injected. How many?
10. Histological sections… How many sections were taken from each biopsy?
11. ..,with images taken with an optical microscope… What was the magnification? How was it chosen?
12. What was the haemotoxin used? Batch, brand? The same for eosin.
13. Sections were then analysed by a veterinary pathologist (GL). How many sections were analysed? Is the entire section or just selected regions? How was this chosen?
14. Samples were examined via microscopy (Leica DMi1) and processed by ImageJ software. How many samples and what was the magnification? Did the authors perform control groups using only primary antibody/secondary antibody? The controls are important to see the antibody specificity.
15. Figure 1. Please include scale bars. Are all the biopsies of the same size?
16. I can not clearly see the difference between untreated and venom-treated groups. It is possible to quantify the colour intensity using Image J? Can the authors use arrows to highlight the mentioned lesions?
17. Figure 2. Please provide the scale bar for fluorescence images.
18. Figure 2. Why was only one region of the entire section assessed?
19. I suggest the quantification of the extent of cell apoptosis and necrosis. Representative images provide a smaller picture of the real scenario, while quantitative data is more informative and can be assessed using statistical analyses.
20. Figure 3 and 4. Please provide a scale bar.
21. Figure 4. A quantitative analysis is important to support the findings.
22. Figure 4. How do the authors differentiate ulceration from artefacts? What are the main features of ulceration? Please provide details.
23. Please clarify the dose used. Antivenom interventions, which where delivered one hour following the subcutaneous injection of E. ocellatus or N. nigricollis venom into biopsies, were evaluated at 48 hours to determine whether antivenom treatments were effective in neutralising venom-induced pathology.
24. Please avoid one-sentence paragraphs. Expand the idea or connect with previous paragraph.
25. Figure 6. A statistical analysis must be performed.
26. Please revise formatting. Different fonts have been used in the manuscript.
27. Is this the first time that viper and elapid venoms have been compared regarding to a local effect? How similar are the venoms in terms of action?
28. Can the authors better justify this: their venoms contain a rich diversity of toxins? I also suggest revising this sentence. Venoms in general are diverse. I do not see this as a reasonable justification. Commonly, vipers are more diverse with more toxin families than elapid venoms.
29. Considering the differences in venom composition between viper and elapid authors need to expand the discussion giving some insights to explain similarities and differences in their findings.
30. A figure comparing viper and elapid venoms based on the findings would be useful. On the other hand, the authors can summarise the main insights that this new model has provided for the venom field.
31. The conclusion must be based on the findings. Although is an interesting tool, the new model has not been compared with conventional approaches. In this sense, there is an important gap in its translation.
32. The limitations of the proposed model should be discussed and included in the discussion section.
Author Response
We would like to thank the reviewer for the careful and thorough reading of this manuscript and for the thoughtful comments and constructive suggestions, which has greatly helped to improve the quality of our manuscript revision. Our detailed responses to each point follow below.
- Abstract section. Please specify the morphological changes and the clinical signs for better understanding.
Author response:
We have provided a more detailed description in the revised manuscript:
Line 24-26: Histological analysis of venom-injected ex vivo human skin biopsies revealed morphological changes in the epidermis (ballooning degeneration, erosion, ulceration) comparable to clinical signs of local envenoming.
- Abstract section. What are the antivenom interventions?
Author response:
We have provided a more detailed explanation in the revised manuscript:
Line 31-35: To investigate the efficacy of antivenom, SAIMR Echis monovalent or SAIMR polyvalent was injected one hour following E. ocellatus or N. nigricollis venom treatment, respectively, and although antivenom did not prevent venom-induced dermal tissue damage, it did reduce all pro-inflammatory chemokines, cytokines and growth factors to normal levels after 48 hours.
- Abstract section. reduce pro-inflammatory marker levels. All of them?
Author response:
We have added the following sentence clarification to the abstract:
Line 31-35: To investigate the efficacy of antivenom, SAIMR Echis monovalent or SAIMR polyvalent was injected one hour following E. ocellatus or N. nigricollis venom treatment, respectively, and although antivenom did not prevent venom-induced dermal tissue damage, it did reduce all pro-inflammatory chemokines, cytokines and growth factors markers to normal levels after 48 hours.
- In the resubmission of the manuscript. please include number lines to facilitate feedback and peer review. Process
Author response:
We thank the reviewer for this point and have now added in line numbers.
- Please include references supporting the following sentence: However, our current knowledge of local envenoming pathogenesis emanates primarily from subjecting murine models to venoms from Latin American snakes.
Author response:
We would like to thank the reviewer for this comment. Following this sentence/statement, we have provided detailed examples that are referenced:
Line 75-78: Our current knowledge of local envenoming pathogenesis emanates primarily from subjecting murine models to venoms from Latin American snakes. For example, an isolated P-I SVMP from Bothrops asper venom induced paw oedema and blister formation, in addition to increases in inflammatory mediators (e.g. IL-1 and IL-6) and leukocyte adhesion molecules [19-22].
- The introduction can be enriched with a parallel between the clinical pathology of viper and elapid snake venoms regarding local envenoming. What are the similarities and differences?
Author response:
We would like to thank the reviewer for this suggestion and have added the following paragraph highlighting the similarities and differences between viper and elapid snake venom pathology regarding local envenoming to the introduction:
Line 55-68: Viperid venoms and venoms from some elapid species, such as spitting cobras, cause rapid and extensive local tissue damage [6,7], as well as local pain or hyperalgesia due to inflammation [8-11]. In viperids, this is primarily due to snake venom metalloproteinases (SVMPs) degrading cell basement membranes at the dermal-epidermal interface, separating the dermis from the epidermis and forming blisters [12]. In addition to basement membrane hydrolysis, SVMPs cause damage to microvasculature and widespread degradation of the extracellular matrix, resulting in haemorrhage, tissue hypoxia, and impaired regeneration [13]. Viperid group II phospholipase A2s (PLA2s) have been found to contribute to the accumulation of fluid in the tissue (oedema) by inducing cell contraction, leading to irreversible cell damage [14]. In elapid venoms, cytotoxic PLA2s and three-finger toxins directly damage cells, disrupting cell membranes and/or inducing pore formation, including lysosome lysis [15-17]. For both viperid and elapid venoms, SVMPs and PLA2 promote pro-inflammatory responses that could mediate venom-induced tissue damage, however the exact contribution of toxins and inflammation to acute and longer-term local tissue lesions remains unclear [18].
- Please revise the following sentence: however, there are currently no alternate models to in vivo testing that accurately recapitulate local envenoming in snakebite victims. Despite the limitations, some studies use cell-based assays or other strategies that provide insights into local damage comparable to clinical manifestations observed in human victims. For example, the following researches (https://www.nature.com/articles/s41467-023-43510-w, https://www.ncbi.nlm.nih.gov/pmc/articles/PMC10850160/ ) have used epidermal keratinocyte, human cells to study myotoxicity, which is an important local manifestation caused by snake venoms. This point is important to discuss, but it worth mentioning different efforts to reduce the number of animals and develop approaches to assess venom toxicity and antivenom efficacy.
Author response:
We thank the reviewer for this valid point, and have added the following text to the revised manuscript:
Line 93-99: Several in vitro assays [39-41] are used to identify and assess venom toxin function and interaction with existing and promising snakebite therapies, this has included cell-based assays to gain better insight into tissue damage relevant to human snakebite victims [42-44]. However, there are still currently no alternate models to in vivo testing that accurately recapitulate the effects of local envenoming with the complete functional, structural, and biological complexity of skin, including local pro-inflammatory responses.
- How was the number of biopsies studied per time point determined? How was the number of samples determined? In terms of statistical analyses, are two biopsies enough?
Author response:
We thank the reviewer for this valid point. Unfortunately, we were limited in the number of biopsies due to ethical and financial constraints, and this work represents the first study of this kind. In response to the reviewer’s point, we have further emphasised the limitation of our small sample size in our discussion:
Line 725-727: Nevertheless, we have been careful to factor the lack of vascularisation into our interpretations of the results gained using this model, as well as our limited sample size.
- Did 84 biopsies include those used to study the efficacy of antivenom? This is not clear. To investigate the efficacy of antivenom in this model, biopsies were first injected. How many?
Author response:
We have now clarified these sample numbers in the revised manuscript:
Line 148-150: Immediately upon arrival, the human skin biopsies for both venom and antivenom intervention experiments (n= 84) were cultured under standard mammalian cell conditions held at air- liquid interface (37 °C, 5% CO2 and maximum humidity).
- Histological sections… How many sections were taken from each biopsy?
Author response:
Two per biopsy/ time point. We have added an experimental overview figure (Figure S1) to provide greater clarity.
- ..,with images taken with an optical microscope… What was the magnification? How was it chosen?
Author response:
We would like to thank the reviewer for this comment, and we have now made these clarifications in Figure 4 and 5 legends of the revised manuscript. Images were taken at 4x to show the overall extent of epidermal loss (erosion/ulceration).
- What was the haemotoxin used? Batch, brand? The same for eosin.
Author response:
We have now provided these details in the revised manuscript ‘Material and methods’ section: Haematoxylin (TCS Biosciences Ltd HD1475) and Eosin Y Stain (TCS Biosciences Ltd, HS250-1L).
- Sections were then analysed by a veterinary pathologist (GL). How many sections were analysed? Is the entire section or just selected regions? How was this chosen?
Author response:
We have now clarified this in the methods:
Line 187-192: The skin biopsies were bisected at the end of the experiment and one half placed in formalin for histological examination. The cut surface was at the center of the biopsy (site of inoculation) and complete transverse sections of this orientation were examined comparing changes seen at different time points to those seen in control experiments, using an optical microscope (Olympus BX43).
- Samples were examined via microscopy (Leica DMi1) and processed by ImageJ software. How many samples and what was the magnification? Did the authors perform control groups using only primary antibody/secondary antibody? The controls are important to see the antibody specificity.
Author response:
Yes, control groups were used. We have added this clarification in the ‘Materials and methods’:
Line 203-204: Biopsy sections, both with and without venom treatments, were examined via microscopy (Leica DMi1) and processed by ImageJ software.
Line 225-227: Biopsy sections, both with and without venom treatments, were examined with a Leica DM5000 microscope and captured images processed with ImageJ software.
- Figure 1. Please include scale bars. Are all the biopsies of the same size?
Author response:
In the revised manuscript we have now included additional details regarding the HypoSkin model.
Line 299-302: The HypoSkin® model, developed as an ex vivo human skin model by Genoskin (France), comprises of 15 mm in diameter and 10 mm thick biopsies consisting of three skin layers (epidermis, dermis, and subcutis) and subcutaneous fat tissue.
- I can not clearly see the difference between untreated and venom-treated groups. It is possible to quantify the colour intensity using Image J? Can the authors use arrows to highlight the mentioned lesions?
Author response:
We would like to thank the reviewer for this suggestion, we have now updated the photos and have added a circle and an arrow to better indicate the visible venom-induced skin alterations. We have also added the following clarifications in the revised manuscript:
Line 292-296: The temporal development of a faint brownish lesion surrounded by a whitish area in the epidermis and/or superficial dermis at seven days (168-hours) was most pronounced in the biopsy injected with B. arietans venom (Figure 1). Both the ‘non-necrotic’ N. haje venom-injected and non-injected control biopsy exhibited an unaltered macroscopic appearance.
- Figure 2. Please provide the scale bar for fluorescence images.
Author response:
We would like to thank the reviewer for spotting this, a scale bar has been added to Figure 2.
- Figure 2. Why was only one region of the entire section assessed?
Author response:
We have assessed all skin layers (Table S2) but we focused on the epidermal and dermal layers as the majority of the apoptotic and necrotic events were detected in the epidermal and dermal region directly adjacent to venom injection.
- I suggest the quantification of the extent of cell apoptosis and necrosis. Representative images provide a smaller picture of the real scenario, while quantitative data is more informative and can be assessed using statistical analyses.
Author response:
We would like to thank the reviewer for this suggestion. We have now quantified fluorescence intensities and have added a new Figure 3 with these measurements and corresponding statistical analyses to the revised manuscript. We have also added another section in the ‘Materials and methods’ for method details relating to the quantification of fluorescence intensities for skin biopsies:
Line 229-235: Fluorescent intensities were measured using ImageJ software (Java-based image processing) for all skin biopsies stained with DAPI and TUNEL. Measurements were applied to the entire image with the area integrated intensity and mean grey value determined for both areas with fluorescence and areas without fluorescence (as background). Corrected total cell florescence (CTCF) was calculated as CTCF = Integrated Density - (Area of selected cell x Mean fluorescence of background reading) and plotted with GraphPad. Significance was determined as P<0.05 via ANOVA with Dunnett’s multiple comparison test.
Figure 3 and 4. Please provide a scale bar.
Author response:
We would like to thank the reviewer for spotting this, scale bars have now been added to these figures, which also have updated figure numbers 4 and 5.
- Figure 4. A quantitative analysis is important to support the findings.
Author response:
We would like to thank the reviewer for this comment. Does the reviewer request quantification of the epithelium loss? This would be, in our opinion, unhelpful as the shape and distribution of the ulceration would vary on different sections, therefore many multiple sections would be required for this type of analysis to be meaningful. The extent of epithelium loss was described as focal or multifocal; there was no description of severity as this is not appropriate in this model.
- Figure 4. How do the authors differentiate ulceration from artefacts? What are the main features of ulceration? Please provide details.
Author response:
Thank you very much for your question. Any sections which were suspected to show artefactual loss of the epidermis were not analysed and replacement sections were prepared and analysed. Where there was ulceration, this was seen in conjunction with other changes (i.e. degeneration of cells in the epidermis at the periphery of the ulceration) and often degeneration of the underlying collagen, changing its appearance, as can be seen in Figure 4B (this has now been updated to be Figure 5)
- Please clarify the dose used. Antivenom interventions, which where delivered one hour following the subcutaneous injection of E. ocellatus or N. nigricollis venom into biopsies, were evaluated at 48 hours to determine whether antivenom treatments were effective in neutralising venom-induced pathology.
Author response:
We have added antivenom concentrations in the revised manuscript, details which are now listed in the ‘Materials and methods’:
Line 163-169: To investigate the efficacy of antivenom in this model, biopsies were first injected subcutaneously with E. ocellatus venom (120 µg) or N. nigricollis venom (165 µg), then one hour later, the SAIMR Echis monovalent (Lot, BC00147, Exp JAN 2016; South African Vaccine Producers; vial concentration 52 mg/ml) or SAIMR polyvalent (Lot, BF0546, Exp Nov 2017 South African Vaccine Producers; vial concentration 103 mg/ml) antivenoms were subcutaneously injected (20 µl of neat antivenom/biopsy) into the E. ocellatus or N. nigricollis ‘envenomed’ biopsy, respectively.
- Please avoid one-sentence paragraphs. Expand the idea or connect with previous paragraph.
Author response:
Thank you, we have now corrected this in the revised manuscript.
- Figure 6. A statistical analysis must be performed.
Author response:
We would like to thank the reviewer for this comment. We have now preformed statistical analyses on this dataset and have assembled a supplemental table (Table S2) with these results. Additionally, we have added p-values to the manuscript text to note significance.
- Please revise formatting. Different fonts have been used in the manuscript.
Author response:
We thank the reviewer for this point and apologise for the formatting. We have now corrected this to improve the revised version of the manuscript.
- Is this the first time that viper and elapid venoms have been compared regarding to a local effect? How similar are the venoms in terms of action?
Author response:
We would like to thank the reviewer for his question and noted significance of our work. As far as we know, there has not been another study that has made this comparison. Therefore, we have generated a summary table from our study (Table S3) to show the similarities and differences we have observed between viper and elapid venoms, and each species, regarding local envenoming. Here we have summarised what we have observed for both histopathology and inflammatory responses.
- Can the authors better justify this: their venoms contain a rich diversity of toxins? I also suggest revising this sentence. Venoms in general are diverse. I do not see this as a reasonable justification. Commonly, vipers are more diverse with more toxin families than elapid venoms.
Author response:
Thank you for this point, we have removed the statement ‘their venoms contain a rich diversity of toxins’ as it is true that all venoms contain a rich diversity of toxins. The corrected sentence still supports our justification as to why we have chosen to investigate these venoms in the ex vivo skin model:
Line 553-555: We used venoms from sub-Saharan African vipers (E. ocellatus and B. arietans) and a spitting cobra (N. nigricollis) as test vehicles because their envenoming is responsible for considerable snakebite morbidity.
- Considering the differences in venom composition between viper and elapid authors need to expand the discussion giving some insights to explain similarities and differences in their findings.
Author response:
The difficulty with making general conclusions between viper and elapid venoms is that extensive venom variation is present within these snake families, and even within a single species. Additionally, certain toxin families are shared between vipers and elapids (i.e. SVMPs and PLA2s), making it difficult to also distinguish in our skin biopsies treated with venom, and not isolated toxins, what toxins are responsible for observed tissue damage and if these toxins are unique to either viperids (i.e. SVMP PII) or elapids (i.e. 3FTxs). Even if we were to identify these toxins it is unknown if their mechanism of action would be the same for all venomous snakes in a family, likely not. However, we have added more information regarding differences in mechanism of action between distinct venom toxin families in vipers and elapids to the introduction:
Line 55-68: Viperid venoms and venoms from some elapid species, such as spitting cobras, cause rapid and extensive local tissue damage [6,7], as well as local pain or hyperalgesia due to inflammation [8-11]. In viperids, this is primarily due to snake venom metalloproteinases (SVMPs) degrading cell basement membranes at the dermal-epidermal interface, separating the dermis from the epidermis and forming blisters [12]. In addition to basement membrane hydrolysis, SVMPs cause damage to microvasculature and widespread degradation of the extracellular matrix, resulting in haemorrhage, tissue hypoxia, and impaired regeneration [13]. Viperid group II phospholipase A2s (PLA2s) have been found to contribute to the accumulation of fluid in the tissue (oedema) by inducing cell contraction, leading to irreversible cell damage [14]. In elapid venoms, cytotoxic PLA2s and three-finger toxins directly damage cells, disrupting cell membranes and/or inducing pore formation, including lysosome lysis [15-17]. For both viperid and elapid venoms, SVMPs and PLA2 promote pro-inflammatory responses that could mediate venom-induced tissue damage, however the exact contribution of toxins and inflammation to acute and longer-term local tissue lesions remains unclear [18].
Another factor that makes it difficult for direct comparisons to be made from our study is that because we based our venom dosing on mouse MND data, different doses of each venom were used in our experimental work. Therefore, it is even difficult to make the conclusion that for instance, Bitis arietans venom causes more local tissue damage (even though we see this in Figure 1, 2, and 3) because we injected 195 μg of B. arietans venom, the greatest amount in comparison to the other venoms.
- A figure comparing viper and elapid venoms based on the findings would be useful. On the other hand, the authors can summarise the main insights that this new model has provided for the venom field.
Author response:
We would like to thank the reviewer for his suggestion. A summary table (Table S2) from our study has been added to the manuscript to show the similarities and differences in terms of morphology changes and pro inflammatory response in the ex vivo human skin model resulting from viper and elapid venom treatments. We have made sure to list these observations for each species, as to avoid a generalised viper and elapid comparison.
The conclusion must be based on the findings. Although is an interesting tool, the new model has not been compared with conventional approaches. In this sense, there is an important gap in its translation.
Author response:
We thank the reviewer for this comment, and we have clarified in the conclusions that there is still additional experimental work required to make this model fully translational:
Line 734-739: Compared to murine models of local envenomation, ex vivo human biopsies demonstrated similar histological signs of epidermal necrosis and apoptosis and were stable over long time periods (up to 168-hours). Once standardized, this model could be used to assess the effectiveness of antivenom and other complementary therapies against venom-induced dermonecrosis.
- The limitations of the proposed model should be discussed and included in the discussion section.
Author response:
We have now added a paragraph to the discussion that addresses the limitations of the model:
Line 718-732: The model occupies the research space between clinical observations and in vivo animal models. An obvious weakness of this model is the inability to mimic intravenous interventions, which, in the field of snakebite therapy, places clear limits on model utility. The need for experimentation within a day of biopsy excision, and the costs of human skin biopsies, will inevitably restrict experimentation to well-funded groups close to surgical facilities. The lack of vascularisation of the ex vivo biopsies might also exert physiological conditions restricting the duration of experiments and the cells/systems that can be validly examined, however, our comparisons between control and venom-injected biopsies in terms of skin morphology, cell death and skin-cell expression of a wide variety of cytokines, chemokines and growth factors identified that the biopsies were physiologically viable and responsive to tissue injury for at least six days after excision. The ability to capture these datasets from human skin tissue therefore represents a significant clinical relevance advancement over the murine models of local envenoming. Nevertheless, we have been careful to factor the lack of vascularisation into our interpretations of the results gained using this model, as well as our limited sample size.
Round 2
Reviewer 3 Report
Comments and Suggestions for Authors
The authors have carefully addressed all the raised points. Their answers were clear and significantly contributed to improving the manuscript. They also recognize the limitations and the reduced number of samples. However, this study is relevant, and their efforts to develop innovative approaches for studying the local damage induced by medically important snake venoms are novel and valuable. In conclusion, I recommend its publication in its current form.
Author Response
Thank you for your supportive comments.